# Health and disease markers correlate with gut microbiome composition across thousands of people

Ohad Manor [1,4✉], Chengzhen L. Dai[2,4], Sergey A. Kornilov [2], Brett Smith[2], Nathan D. Price [2,3], Jennifer C. Lovejoy[2], Sean M. Gibbons [2,3] & Andrew T. Magis[2]

Variation in the human gut microbiome can reflect host lifestyle and behaviors and influence disease biomarker levels in the blood. Understanding the relationships between gut microbes and host phenotypes are critical for understanding wellness and disease. Here, we examine associations between the gut microbiota and ~150 host phenotypic features across ~3,400 individuals. We identify major axes of taxonomic variance in the gut and a putative diversity maximum along the Firmicutes-to-Bacteroidetes axis. Our analyses reveal both known and unknown associations between microbiome composition and host clinical markers and life-style factors, including host-microbe associations that are composition-specific. These results suggest potential opportunities for targeted interventions that alter the composition of the microbiome to improve host health. By uncovering the interrelationships between host diet and lifestyle factors, clinical blood markers, and the human gut microbiome at the population-scale, our results serve as a roadmap for future studies on host-microbe interactions and interventions.

---

[1] Century Therapeutics, Seattle, WA 98102, USA. [2] Institute for Systems Biology, Seattle, WA 98109, USA. [3] Department of Bioengineering, University of Washington, Seattle, WA 98105, USA. [4]These authors contributed equally: Ohad Manor, Chengzhen L. Dai. ✉email: omanor@gmail.com

The human gut microbiome—the collection of microorganisms residing in the gastrointestinal tract—is thought to play a role in the etiology of various diseases, including inflammatory bowel disease[1–3], type 2 diabetes[4–6], hypertension[7–9], and colorectal cancer[10–13]. Individual clinical blood markers, such as those for diabetes[14] and cholesterol[15], have been found to be associated with abundances of certain gut bacteria. Lifestyle habits can also impact the composition of the gut microbial community. For example, diet can profoundly influence the composition of the microbiome[16–18]. Similarly, physical activity has been shown to drive shifts in the composition of the gut microbiome in animal models[19,20], and there is preliminary evidence from small-cohort studies that exercise impacts the microbiome in humans as well[21–25].

Despite the growth in microbiome research in recent years, large-scale human studies that integrate gut microbiome profiles with host clinical blood phenotypes, dietary, and lifestyle data, and disease and medication usage remain scarce. These dense phenotyping studies on large cohorts are crucial for validating associations established in varying contexts, using sparser data from smaller cohorts.

Here, we provide an in-depth analysis of the relationship between the gut microbiome and host factors across ~3400 healthy US individuals in a large cross-sectional study. We identify lifestyle and clinical factors, including diet, medication use, and clinical blood markers, that are associated with the composition of the gut microbiome, including diversity, individual taxonomies, and inferred functional pathways. Stratifying individuals by the major axes of variation in microbial composition, we identify lifestyle behaviors and host factors that are associated with microbiome diversity only in certain underlying microbiome contexts. Finally, we identify host-microbiome associations that are robust and independent from the underlying species diversity of the microbial community.

## Results

**Characteristics of study participants**. The data presented in this study were collected at baseline from 3409 research-consenting participants in a commercial Scientific Wellness program (Arivale Inc., see Methods). Participants included 59% females, 84% self-reported European-Americans, with an average age of $49 \pm 12$ years, and an average BMI of $27 \pm 6 \, kg/m^2$ (Table 1). All participants completed lifestyle, stress, digestion, and diet questionnaires, and for 90% ($n = 3064$) of participants, 65 clinical laboratory tests were measured from blood (Supplementary Data 1). Gut microbiome composition from a baseline stool sample was determined by 16S amplicon sequencing for all participants (see Methods).

**Microbiome diversity is strongly associated with the Bacteroidetes-to-Firmicutes axis**. When examining the overall composition of the gut microbiome in the study participants, the relative abundance of Firmicutes ranged from ~6% to ~100%, while the relative abundance of Bacteroidetes ranged from ~0% to ~90% (Fig. 1). We observed that Shannon diversity was strongly

negatively correlated with the relative abundance of Bacteroidetes ($r = -0.67$, $P < 10^{-15}$, Pearson's correlation), however, this trend was not linear and showed a positive correlation for the lowest levels of Bacteroidetes. This nonlinear trend was also observed in other measures of diversity such as Pielou's evenness index, species richness, and Faith's phylogenetic index (See Supplementary Fig. 1). Overall, Shannon diversity was maximized when the relative abundances of Bacteroidetes and Firmicutes were ~15% and ~80%, respectively (Supplementary Fig. 2; see Methods). In addition, the phyla Proteobacteria, Fusobacteria, and TM-7 were significantly negatively correlated with diversity ($r = -0.18$, $P < 10^{-15}$; $r = -0.13$, $P < 10^{-13}$; $r = -0.10$, $P < 10^{-8}$, respectively), while the phyla Tenericutes, Euryarchaeota, Lentisphaerae, and Cyanobacteria were significantly positively correlated with diversity ($r = 0.28$, $P < 10^{-15}$; $r = 0.19$, $P < 10^{-15}$; $r = 0.17$, $P < 10^{-15}$; $r = 0.13$, $P < 10^{-13}$, respectively).

To identify correlation patterns between taxa and diversity that go beyond the relative abundance of Bacteroidetes (and since Bacteroidetes was the phylum most correlated with diversity), we re-examined the correlations between taxa and diversity while accounting for Bacteroidetes abundance. To this end, we defined Bacteroidetes-adjusted diversity (BA-diversity) as the residuals from a nonlinear regression of diversity on Bacteroidetes abundance (see Methods). We found that the phylum Actinobacteria, which was slightly positively correlated with unadjusted Shannon diversity ($r = 0.04$, $P = 0.02$), was the most negatively correlated phylum with BA-diversity ($r = -0.21$, $P < 10^{-15}$), and its two genera *Bifidobacterium* and *Eggerthella* were among the most negatively correlated genera with BA-diversity ($r = -0.24$, $P < 10^{-15}$; $r = -0.20$, $P < 10^{-15}$, respectively). Notably, while *Bifidobacterium* is considered a beneficial gut-dwelling bacterial genus[26], *Eggerthella* (and specifically, *E. lenta*) has been described as an opportunistic pathogen[27].

To examine the main drivers of taxonomic variance in the gut microbiome of participants, we applied edgePCA[28], a microbiome-specific principal component analysis that accounts for phylogeny (Fig. 2). The first taxonomy-based principal component (PC1; explaining ~54% of variance) was found to be strongly correlated with the abundances of the phyla Firmicutes and Bacteroidetes (Fig. 2b), whereas the second principal component (PC2; explaining ~17% of variance) was found to be strongly correlated with the abundances of the Bacteroidetes genera *Prevotella* and *Bacteroides* (Fig. 2c). We however did not find distinct clusters as previously reported[29], but rather a continuum of compositions ranging from very low levels of overall Bacteroidetes, to high levels of either *Prevotella* or *Bacteroides* (Fig. 2a). As reported in previous studies[18,29], *Prevotella*- and *Bacteroides*-rich compositions were found to be relatively non-overlapping.

Extending our analysis to the third and fourth principal components (PC3-4; explaining ~5% and ~3% of variance, respectively), we found that different clades from the order Clostridiales (phylum Firmicutes) were strongly correlated with these PCs (Supplementary Fig. 3). Specifically, the genus *Faecalibacterium* (family *Clostridiaceae*) was negatively correlated with PC3, while the families *Lachnospiraceae* and *Ruminococcaceae* were negatively and positively correlated with PC4, respectively. These results indicate that among the continuum of increased Firmicutes abundance, compositions tend to vary in these three clades. Interestingly, the Actinobacteria genus *Bifidobacterium*, known to be a beneficial commensal genus[26,30], was found to be positively correlated with PC3 (and therefore negatively correlated with *Faecalibacterium*, another beneficial microbial genus[31,32]).

We also compared these results with the results obtained by applying nonmetric multidimensional scaling (NMDS) to either

| Table 1 Summary of demographics in individuals. | |
|---|---|
| **Feature** | **Overall** |
| Number of individuals | 3409 |
| Sex (% female) | 59% |
| Age | $49 \pm 12$ |
| Ethnicity (% white) | 84% |
| BMI | $27 \pm 6$ |

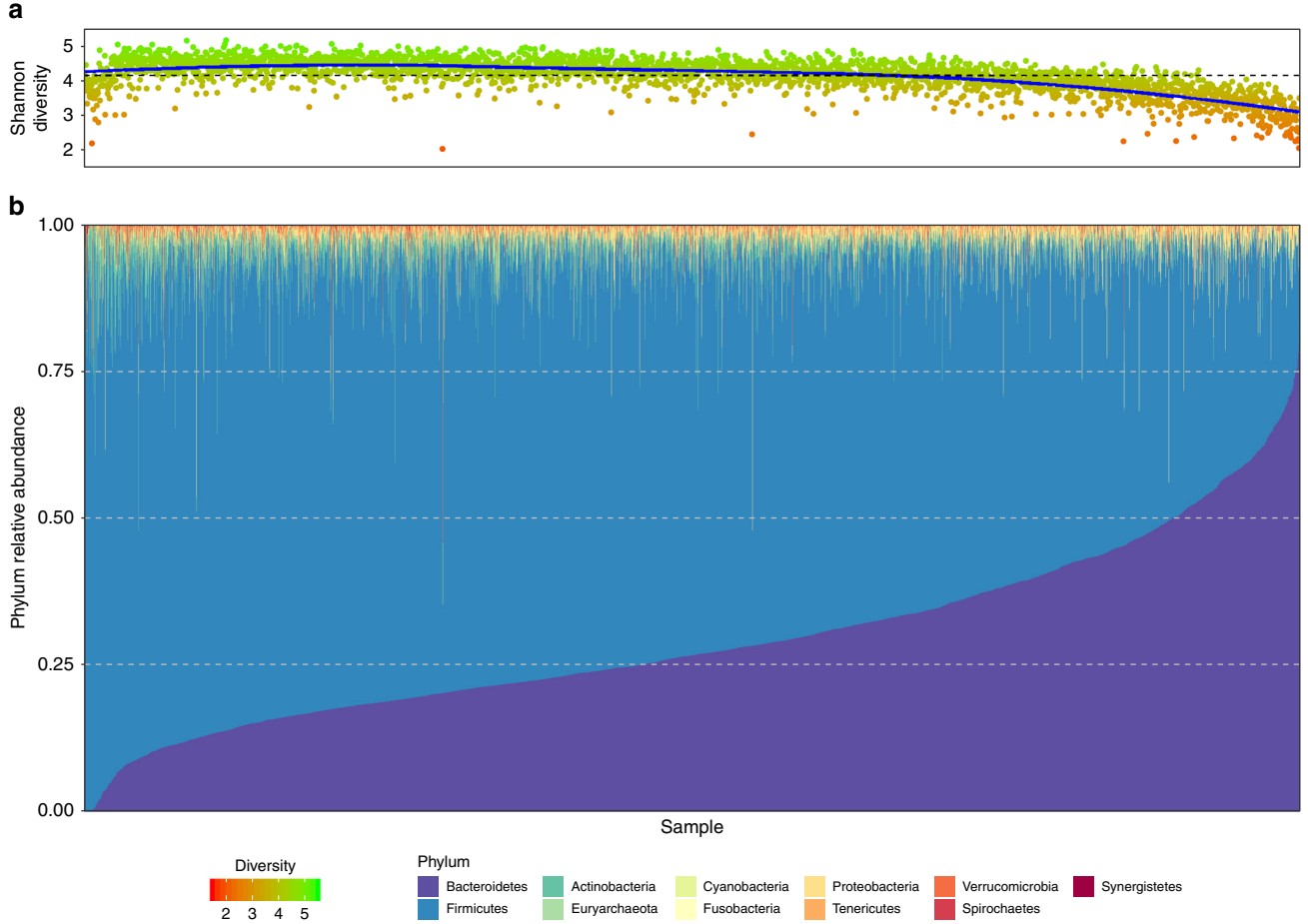

**Fig. 1 Overall composition of the gut microbiome in participants.** Shown are the Shannon diversity index (**a**) and phyla relative abundances (**b**) across all participants' samples. Each sample is represented by one stacked bar in the bottom panel (**b**) colored by phyla, and a corresponding point in the upper panel (**a**) indicating the microbial diversity.

weighted UniFrac or Bray-Curtis distances and found similar results, although the principal components of edgePCA were better correlated with these main taxonomic drivers than the principal coordinates of Bray-Curtis or UniFrac-based NMDS, highlighting the benefit of using a microbiome-specific ordination technique (see Supplementary Figs. 4-5).

**Multiple lifestyle and clinical factors are associated with gut microbiome diversity.** We next set out to identify, which host factors are associated with the diversity of the gut microbiome. Out of 148 lifestyle and clinical factors we examined, 75 were significantly correlated with Shannon diversity (Fig. 3 and Supplementary Data 2; FDR-adjusted $P < 0.05$, see Methods). The identified 42 blood clinical markers included some markers previously reported to be associated with microbiome diversity such as markers for diabetes (fasting insulin, $P < 10^{-12}$), inflammation (hs-CRP, $P < 10^{-14}$), liver function (ALAT, $P < 10^{-9}$), and cholesterol (LDL, $P < 10^{-16}$). In addition, omega-3 fatty acids and other markers related to fish intake, such as DHA and mercury, were found to be positively correlated with diversity ($P < 10^{-13}$; $P < 10^{-14}$; $P < 10^{-16}$; respectively). BMI, weight, and blood pressure were significantly negatively correlated with diversity, while height was found to be significantly positively correlated with diversity.

For measured lifestyle factors, we found strong evidence for associations between physical activity and microbiome diversity,

with both the frequency (i.e., number of days per week) and duration of physical activity positively correlated with microbiome diversity. Eating more servings of fruits, vegetables, and cruciferous vegetables was also positively correlated with diversity, while increased consumption of sugary drinks was negatively correlated with diversity. Lastly, indicators of poor bowel health, such as the weekly frequency of diarrhea, nausea, and acid reflux, were negatively associated with microbiome diversity (see Supplementary Data 2 for a full list of association results).

Recent animal studies and small-scale case-control human studies have indicated a relationship between exercise and the gut microbiome composition[19,20,22,25,33], but the robustness of this association at population-level remains uncertain. Here we identify associations between microbiome diversity and both moderate physical activity (MPA; times per week) and vigorous physical activity (VPA; times per week) that are highly significant (Fig. 3; $P < 10^{-15}$ and $P < 10^{-15}$, for MPA and VPA, respectively). Since many host factors are co-correlated, we sought to understand whether the relationship with physical activity is independent of other healthy lifestyle habits. We thus adjusted for related dietary factors, such as weekly intake of fruits, vegetables, whole grains, and sugary drinks, and found that the association remained significant ($P < 10^{-5}$ and $P < 10^{-15}$, for MPA and VPA, respectively). When BMI was included as an additional covariate, the correlation for vigorous physical activity remained significant ($P = 0.08$ and $P = 0.04$, for MPA and VPA,

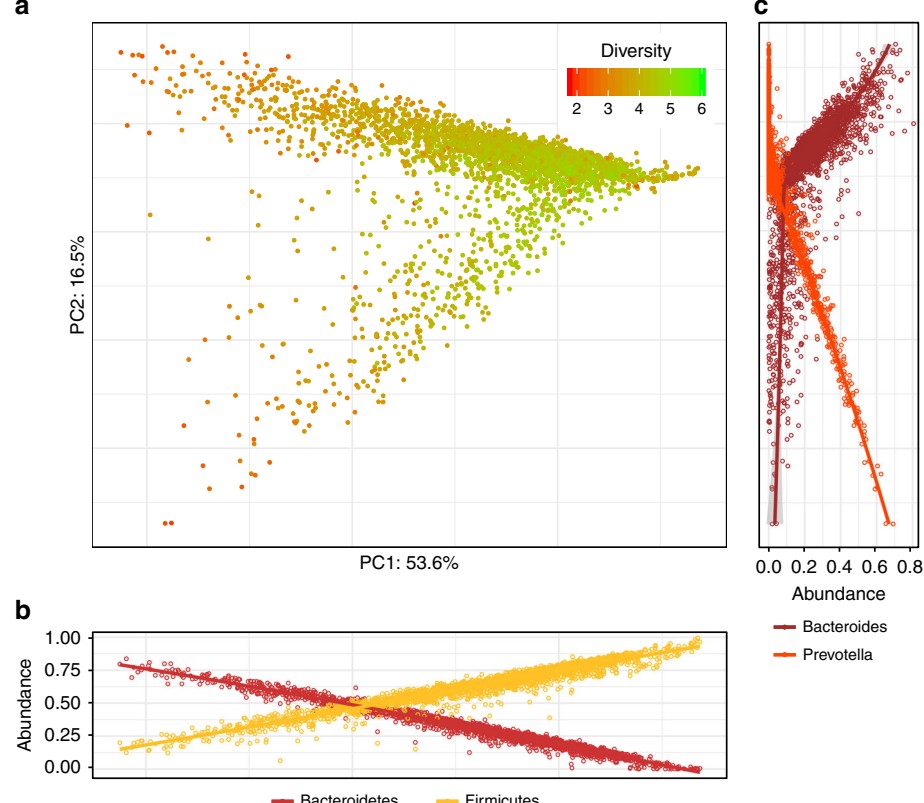

**Fig. 2 Principal component analysis of the taxonomic composition of the gut microbiome. a** Shown is the principal component analysis (PCA) plot generated by applying edgePCA[28] to the OTU-level counts data. Each point represents one sample, and the samples are colored by their Shannon diversity index. The percent of variation explained by each principal component is depicted on each axis. **b** + **c** Shown are scatter plots of the relative abundances of the phyla Bacteroidetes and Firmicutes (**b**) and the genera *Bacteroides* and *Prevotella* (**c**) across the corresponding principal component. For each sample in (**a**), there are two corresponding points in (**b**) and two corresponding points in (**c**). Lines indicate the loess regression fit and the shaded area represents the 95% confidence interval. $N = 3409$ biologically independent samples in all panels.

respectively). Our results thus reveal a robust relationship at population-level between physical activity and microbiome diversity that is independent of major dietary factors and BMI.

**Associations between host factors and microbiome diversity within different compositional clusters.** To understand whether the associations between microbiome diversity and host factors differ in the context of different microbiome compositional states, we defined four clusters on the continuum of Firmicutes-to-Bacteroidetes and *Bacteroides*-to-*Prevotella* axes: a reference cluster with an average Firmicutes relative abundance level, a Firmicutes-rich cluster, a *Bacteroides*-rich cluster, and a *Prevotella*-rich cluster (see Methods and Supplementary Fig. 6). Compared to the associations found within the average Firmicutes cluster (as the reference group), we identified 0, 14, and 14 unique associations within the Firmicutes-rich, *Bacteroides*-rich, and *Prevotella*-rich clusters, respectively ($p < 0.05$ after FDR correction, Supplementary Data 3). For example, we found that the association between insulin levels and microbiome diversity was significantly more pronounced in the *Bacteroides*-rich cluster than the reference cluster ($p < 10^{-6}$) and that the number of vegetables consumed per day was significantly more positively associated with microbiome diversity for the *Prevotella*-rich cluster than other microbiome composition clusters ($p < 10^{-4}$, Fig. 4).

**Host factors aggregate into health- and disease-related groups by their pattern of association with the microbiome.** We next explored the relationship between host factors and individual genera and inferred functional pathways. After FDR correction, we identified >1500 significant associations across 142 host factors, 102 bacterial genera, and 274 bacterial metabolic pathways (see Methods). We found that host factors aggregate into two large groups by their patterns of microbiome associations (Fig. 5, see full list of associations in Supplementary Data 4). The first is a health-related group, where higher values of the host factors or increased cadence of lifestyle behaviors are generally associated with better overall health, including clinical lab measurements (e.g., vitamin D, HDL, LDL particle size, omega−3 index, and adiponectin); dietary factors (e.g., consumption of fruits and vegetables); activity factors (e.g., physical activity); and digestion factors (e.g., bowel movement ease). The second is a disease-related group, in which higher values of the host factors or increased cadence of lifestyle behaviors are generally associated with worse overall health, including BMI; diabetes markers (e.g., HOMA − IR, Insulin, glucose, HbA1c); cardiovascular risk factors (e.g., LDL cholesterol, triglycerides, and blood pressure); inflammation risk markers (e.g., CRP − HS, IL − 6); and poor digestion symptoms (e.g., diarrhea, acid reflux). Each group was highly associated with a different set of genera. The factors in the health-related group were positively correlated and significantly explained the abundance of the genera *Coprococcus*, *Lachnospira*, *Faecalibacterium*, and unclassified genera from the families/orders Ruminococcaceae, Ml615j − 28, Clostridiales, and Rf39. Some of these genera were previously reported to be health-promoting; for example, *Lachnospira* and *Faecalibacterium* are known producers of the anti-inflammatory short-chain fatty acid

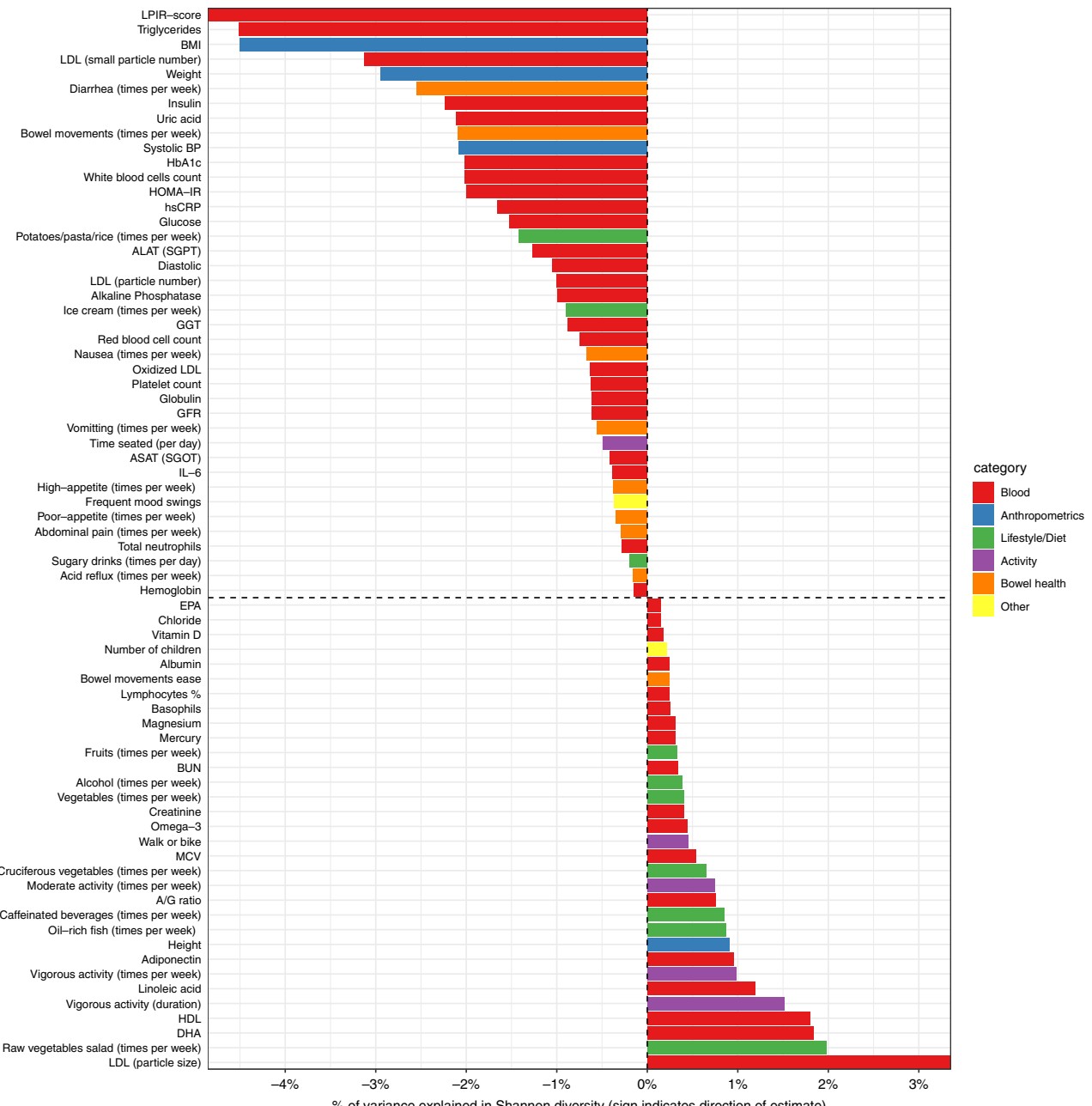

**Fig. 3 Significant associations between microbiome diversity and multiple factors.** Shown is a bar plot representing the percent of variance explained (x-axis) in Shannon diversity across all available samples by each factor (y-axis). Positive values indicate positive associations while negative values indicate negative associations with diversity. Each factor is colored by the category to which it belongs. For each analyte (e.g., lifestyle, diet, clinical test), associations were tested by fitting linear regression models (see Methods) and only factors that passed the FDR[49] multiple hypothesis correction with $p < 0.05$ are shown.

butyrate. The factors in the disease-related group significantly explained (and were positively correlated with) the abundance of the genera *Bacteroides*, *Ruminococcus*, *Sutterella*, *Bilophila*, *Acidaminococcus*, and *Megasphaera*.

To understand whether the associations with individual taxa represent independent associations or reflect a more global compositional shift in the microbiome, we repeated the analysis described above but this time also adjusting for Shannon diversity in our models. We found that *Ruminococcaceae* and *Clostridiales* were both strongly associated with the majority of factors in the health- and disease-related groups prior to adjusting for diversity, but when adjusting for diversity, all *Ruminococcaceae* associations became non-significant, while almost all *Clostridiales* associations

remained significant (Fig. 5 and Supplementary Fig. 7). In addition, the genera *Bacteroides* and *Sutterella* showed a similar pattern of significant associations with host and lifestyle factors, but when we adjusted for Shannon diversity, most *Bacteroides* associations became non-significant, while *Sutterella* associations remained significant (Fig. 5 and Supplementary Fig. 7). These results suggest that for *Ruminococcaceae* and *Bacteroides*, associations with host factors might result indirectly from the correlations between these genera and diversity, while *Clostridiales* and *Sutterella* show the same patterns of associations for individuals with similar Shannon diversity values.

We also tested whether there are factor-taxon associations that were different across the four Firmicutes-*Bacteroides*-*Prevotella*

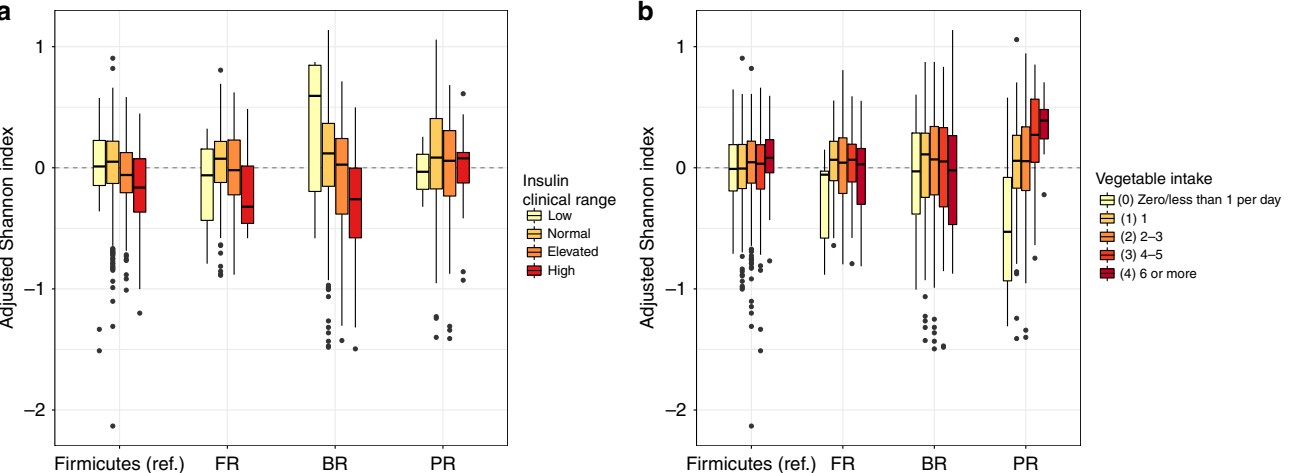

**Fig. 4 Host factors show different patterns of association with microbiome diversity across taxonomic clusters.** Shown are boxplots of Shannon diversity (adjusted for confounding factors) for clinical ranges of fasting insulin (**a**; N = 2754) and daily intake of vegetables (**b**; N = 2583) across the reference cluster (Firmicutes), and the Firmicutes-rich (FR), *Bacteroides*-rich (BR), and *Prevotella*-rich (PR) clusters. Boxes denote the interquartile range (IQR) between the first and third quartiles with the black line inside each box denoting the median. Whiskers extend to the lowest and highest values within 1.5 times IQR.

composition clusters, repeating the analysis above with the addition of an interaction term between the host factor and microbiome cluster. We found 163 cluster-specific associations (Supplementary Data 5) between host factors and taxa. In the *Prevotella*-rich cluster, we found, for example, that the abundance of an unclassified genus from the family Erysipelotrichaceae was significantly correlated with self-reported bloating and nausea. However, no associations with bloating and nausea were observed for the other microbiome clusters. In addition, we found that the abundance of the genus *Akkermansia* and an unclassified genus from the family *Christensenellaceae* were negatively correlated ($p < 0.05$ FDR) with consumption of cookies and pastries in the *Bacteroides*-rich cluster compared to other clusters (non-significant for other clusters, see Supplementary Data 5 for full list).

When examining the results from the analysis of inferred functional pathways (Methods), we found that the host factors similarly aggregated into a health-related group and a disease-related group based on their association with microbiome pathways (Fig. 6, full list of associations in Supplementary Data 6). The health-related group included host factors and behaviors generally associated with good overall health, such as more physical activity, healthier dietary choices, and better clinical blood chemistries, while factors in the disease-related group were generally associated with poorer overall health, such as increased levels of BMI, digestive symptoms, and risk markers related to diabetes, cardiovascular disease, and inflammation. Notably, most glycan and carbohydrate metabolism pathways were positively associated with the disease-related group. In addition, primary and secondary bile acid metabolic pathways, and vitamin or vitamin-like metabolic pathways were positively associated with the disease-related group. In contrast, xenobiotics metabolism pathways were positively associated with the health-related group. Finally, we adjusted our models to identify associations that are independent of microbiome diversity. We found that certain microbial pathways, such as lipopolysaccharide biosynthesis, were not associated with the disease-related group once diversity was accounted for, while other pathways, such as primary and secondary bile acid metabolism, were still associated with the disease-related group even after adjusting for diversity, indicating a robust association between these microbial pathways and host factors and behaviors.

**Medications impact the abundance of genera and functional pathways in the gut microbiome.** Recent studies have revealed the impact of medications on the gut microbiome[27,34]. We built on these analyses by accounting for co-occurrence between medications and their associated disease by adjusting our models for the respective clinical measurement that serves as biomarkers of disease (Methods). We compared the relative abundance of genera and functions in medication-users vs. non-users and identified 70 significant associations between medication usage and the relative abundance of genera (Supplementary Data 7). The genus *Klebsiella* (an opportunistic pathogen from the family *Enterobacteriaceae*) was found to be significantly more abundant in individuals taking blood-sugar medications (fold-change > 6, $p$ value $< 10^{-4}$), consistent with a recent study of the impact of metformin (a common blood-sugar medication) on the microbiome[35]. On the other hand, the genus *Faecalibacterium* was significantly lower in individuals taking blood-sugar medications (FDR-corrected $P < 10^{-6}$). In addition, members of *Enterobacteriaceae* and Burkholderiales were increased in abundance in individuals using cholesterol-lowering drugs, consistent with a recent study that included an analysis of the impact of statins on the microbiome[36].

When examining the functional composition of the microbiome, we identified 109 and 271 significant associations between medication usage and the predicted abundance of pathways and modules, respectively (Supplementary Data 8–9). Examples of functional shifts that were seen for medication users (adjusted for the respective levels of relevant blood markers) included the enrichment of the metabolic pathway for fructose and mannose metabolism for users of blood-sugar lowering medications, recently shown to be enriched in type 2 diabetes patients using the medication metformin[34].

## Discussion

Host lifestyle and diet can greatly impact the gut microbiome, which in turn can influence host metabolism and wellbeing. Our study of the human gut microbiome across ~3400 individuals with extensive phenotyping of ~150 host factors enabled us to identify host-microbiome associations relevant to health and disease. We also reveal how certain host phenotypic associations

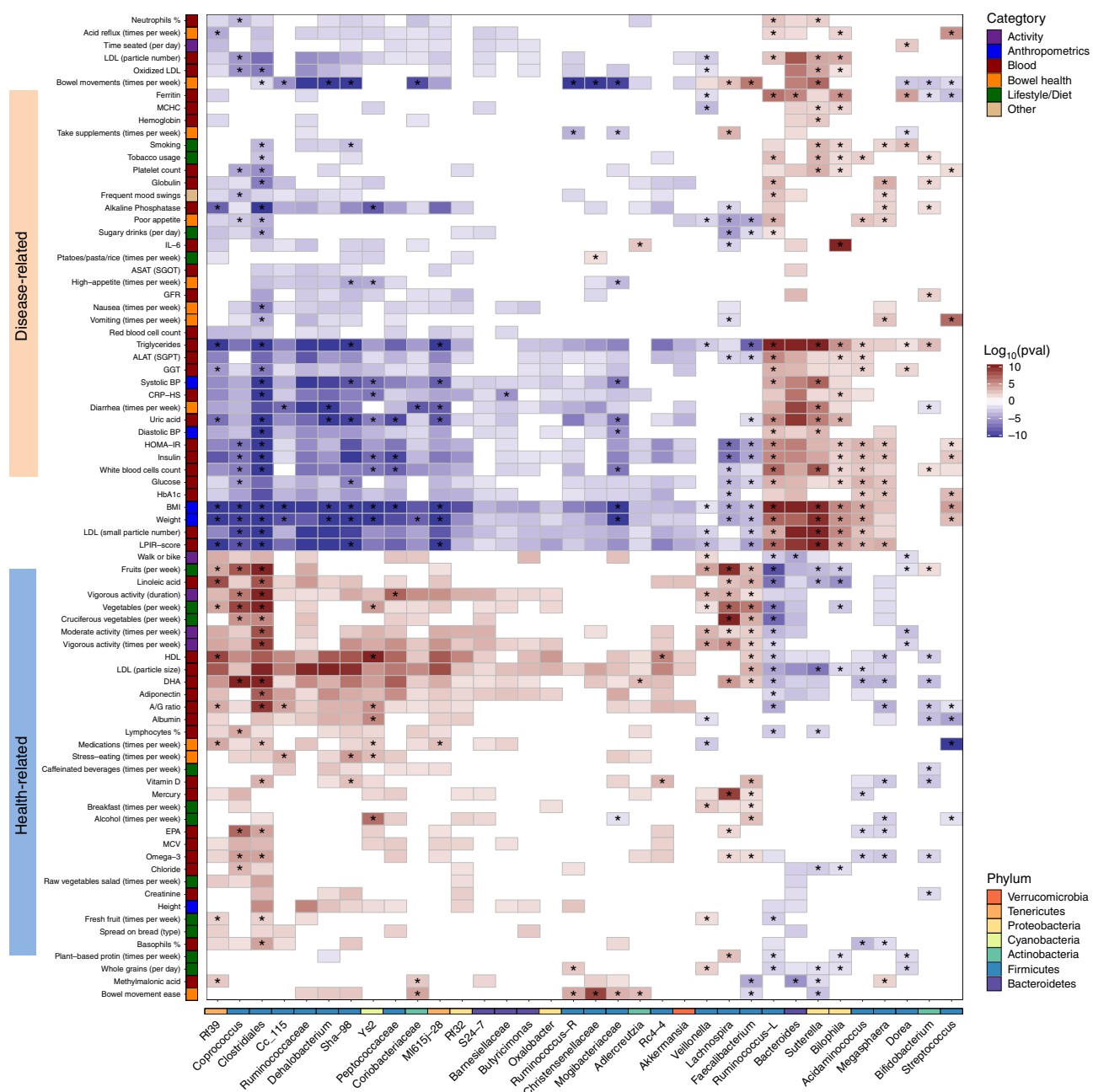

**Fig. 5 Significant associations between microbial genera and multiple factors.** Shown is a heatmap of the microbial genera (*x*-axis) that were found to be significantly associated with different factors (*y*-axis) using generalized linear models adjusted for confounding factors (see Methods). Only genera that had at least 20 significant associations are shown in the plot, along with factors that had at least ten associations with them (see full list of associations in Supplementary Data 3). For each analyte (e.g., lifestyle, diet, clinical test), associations were tested by fitting generalized linear models (see Methods). The significant heatmap cells (after correcting for multiple hypotheses with FDR-corrected *p* < 0.05) are represented by the significance of the *p* value (indicated by saturation, e.g., values of 10 or −10 indicate that *p* value = $1^{-10}$) and the direction of association (indicated by color, e.g., red is positively associated). Each factor is colored by the category to which it belongs, and each genus is colored by the phylum to which it belongs. Associations that are still significant after adjusting for microbiome diversity are marked with an asterisk.

are unique to specific microbial community contexts, suggesting that microbiome heterogeneity may play a role in personalized phenotypic responses to dietary and lifestyle interventions. Overall, our integrated approach provided insights into the complex relationships between the gut microbiome and host phenotypic features, which have implications for future clinical and interventional studies.

Gut microbiome alpha-diversity has been linked to human health, with lower levels of diversity associated with several acute

and chronic diseases[37]. Understanding the underlying taxonomic drivers of microbiome diversity may provide valuable insights into the interaction between our commensal microbiota and our health. Previous analyses of gut community composition and alpha-diversity have primarily used linear approaches or binary comparisons between low and high diversity clusters[38] and have generally indicated a negative association between the Bacteroidetes phylum and diversity. However, gut microbiome comparisons between rural African individuals and urban Europeans

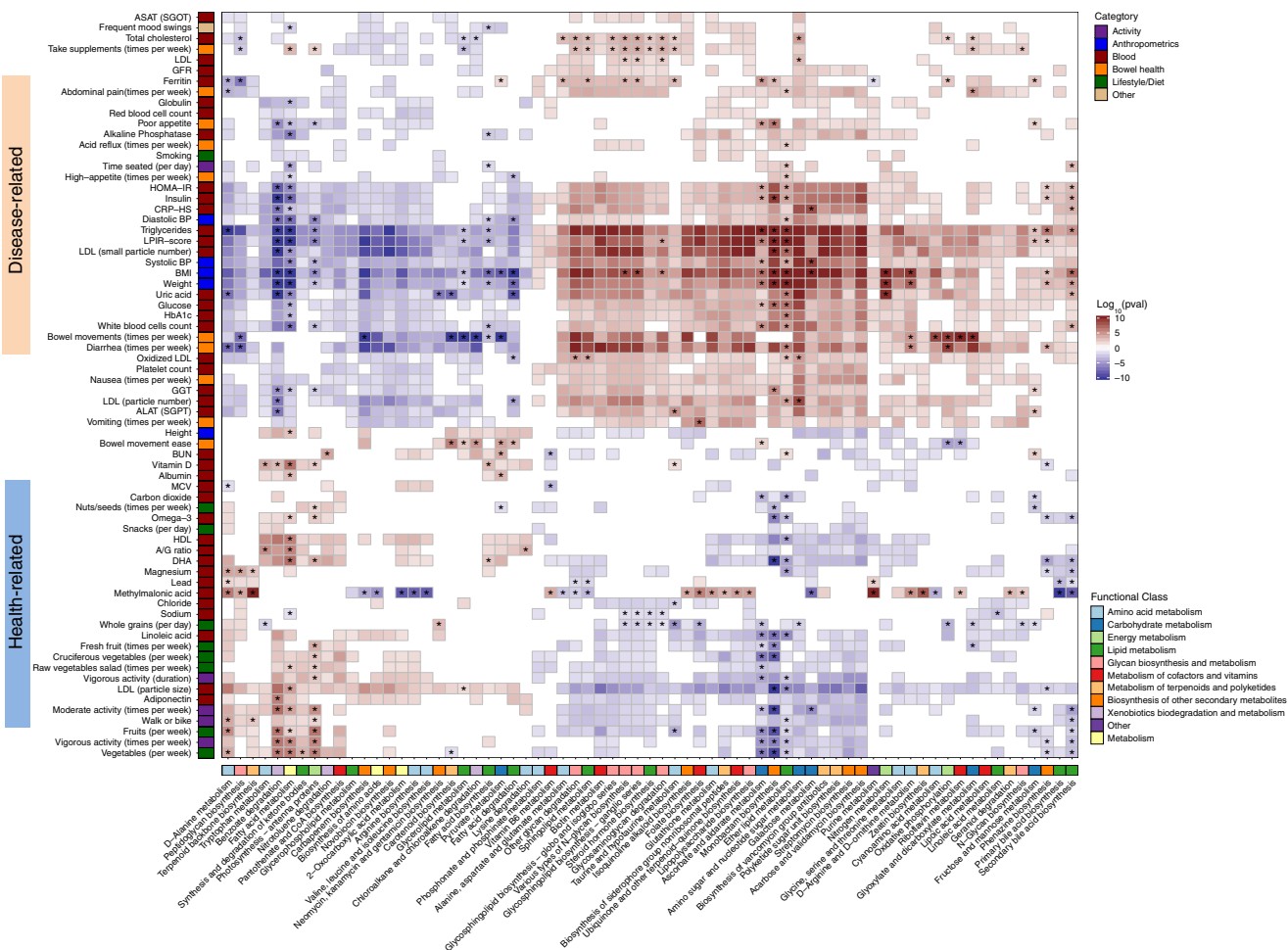

**Fig. 6 Significant associations between microbial pathways and multiple factors.** Shown is a heatmap of the microbial pathways (x-axis) that were found to be significantly associated with different factors (y-axis) using generalized linear models adjusted for confounding factors (see Methods). Only metabolic pathways that had at least 20 significant associations are shown in the plot, along with factors that had at least 15 associations with them (see full list of associations in Supplementary Data 4). For each analyte (e.g., lifestyle, diet, clinical test), associations were tested by fitting generalized linear models (see Methods). The significant heatmap cells (after correcting for multiple hypotheses with FDR-corrected p < 0.05) are represented by the significance of the p value (indicated by saturation, e.g., values of 10 or −10 indicate that p value = 1⁻¹⁰) and the direction of association (indicated by color, e.g., red is positively associated). Each factor is colored by the category to which it belongs, and each genus is colored by the phylum to which it belongs. Associations that are still significant after adjusting for microbiome diversity are marked with an asterisk.

found higher diversity and corresponding enrichment of Bacteroidetes in rural Africans[39,40]. Our finding of a strong, nonlinear association between Bacteroidetes and microbiome diversity suggests that the relationship between these two features is non-monotonic. Thus, a single measure, such as the Firmicutes-to-Bacteroidetes ratio[41], may not comprehensively account for the relationship between composition, diversity, and overall health[42]. Controlling for the abundance of prevalent taxa can reveal some of the subtler associations between microbiome composition and alpha-diversity. The negative association between *Eggerthella*, a putative opportunistic pathogen[27], and Bacteroidetes-adjusted diversity highlights how adjusting for dominant taxa can reveal important underlying associations[43,44] and suggests that deviations from the expected diversity given the underlying composition might serve as an additional biomarker of gut health.

Consistent with prior work, we found that the major axes of gut microbiome variation were the Firmicutes-to-Bacteroidetes axis and the nonoverlapping distributions of the genera *Bacteroides* and *Prevotella*[18,29]. In addition, we found that two putatively beneficial genera, *Bifidobacterium* and *Faecalibacterium*, were negatively correlated with one another. Previous studies

have shown that both of these genera can respond to similar diets (e.g., high fiber) or prebiotics[45]. The inverse relationship suggests that when one of these genera is dominant, the other genus may be out-competed, potentially due to metabolic niche overlap. Similarly, the two Firmicutes families *Lachnospiraceae* and *Ruminococcaceae* are anti-correlated and drive the variation in principal component space. Both families are considered beneficial fiber degraders and butyrate producers[46]. These observations emphasize that there might not exist a single "healthy" microbiome state, but instead, a spectrum or multiple states marked by different sets of beneficial bacteria.

Identifying associations between host lifestyle factors, host health, and the gut microbiome can enable the design of interventions and clinical trials[38,47,48]. Previous studies of these associations, however, have largely focused on overall microbiome variability[38,49]. Our analysis of compositional clusters and cluster-specific associations reveal how the enrichment of certain taxonomies can affect the host-microbiome relationship. These cluster-specific associations have implications for targeted dietary intervention and microbiome modification. For example, frequency of vegetable consumption was positively correlated with

alpha-diversity, but only in the *Prevotella*-rich cluster, suggesting that individuals in that cluster may respond better to plant-rich diets. Similarly, the negative correlation between *Christensenellaceae* and consumption of cookies and pastries in the *Bacteroides*-rich cluster suggest that individuals in that cluster may benefit more from decreasing consumptions of baked goods, particularly since *Christensenellaceae* has been associated with leanness and was shown to be enriched after diet-induced weight loss[50–54]. As clinical trials begin to incorporate microbiome sequencing[55,56], our results highlight the potential of using the microbiome to guide personalized dietary interventions and the value of examining the microbiome composition of participants undergoing dietary modifications to better understand responses to an intervention.

In recent years, many studies have indicated that specific features in the gut microbiome are associated with different health markers, reporting on many pairwise correlations between specific genera in the gut microbiome and different lifestyle or dietary factors[38,49]. However, whether an emergent picture of overall health and wellbeing can be seen in the overall gut microbiome composition, not just with individual taxa, is still an open question. In this study, we not only identified >1500 significant association between host factors and microbiome genera and pathways but also demonstrated that these host factors were grouped according to their *overall* pattern of association across all microbial genera. In particular, host factors aggregated into "health-related" and "disease-related" groups. In fact, many host factors that were not directly correlated across individuals, e.g., HDL cholesterol and the frequency of walking or biking ($p$ value = 0.82; Spearman's correlation), still grouped together in the health-related group when we considered their pattern of microbial associations. Similarly, glycohemoglobin A1c (HbA1c) and frequency of diarrhea—both indicators of poorer health—were not directly correlated across individuals in the population ($P$ value = 0.74, Spearman's correlation) but grouped with factors in the disease-related group based on their microbiome associations. Thus, although certain individual health factors, lifestyle, or dietary choices may not clearly correlate with each other across a population, the underlying co-occurring changes in the gut microbiome associated with these factors do reveal their relationship with health and wellbeing. Further studies of the mediating[57] or moderating[58,59] role of the microbiome in health is therefore needed.

Due to the interrelated nature of many dietary, lifestyle, and clinical factors, independently robust host-microbiome associations have been difficult to uncover. Our large sample size coupled with our extensive phenotyping, however, enabled us to disentangle confounding factors (e.g., BMI, diet) and identify robust associations. We were able to identify a significant independent association between microbiome alpha-diversity and vigorous physical activity, by adjusting for age, sex, BMI, diet, race, and season. The genus *Veillonella* was also positively associated with vigorous physical activity after adjusting for these covariates ($P < 10^{-4}$), consistent with a recent small study of elite marathon runners that found *Veillonella* to be enriched in the guts of athletes[60]. Thus, our results provide evidence that increased vigorous physical activity is independently linked to consistent shifts in the structure and composition of the gut microbiome.

While our extensive phenotyping of host factors and our large sample size enables us to have the statistical power to identify many host-microbiome associations, we note that these associations do not imply causality. Understanding directional and causal relationships will require randomized control studies and interventions, which can control for both known and potential hidden confounders. Similarly, while we adjust for many co-varying factors, we note that statistical adjustment cannot fully account for all confounders, as there are many factors that remain unmeasured. For example, certain health-related activities are manifestations of lifestyle factors but the effect of lifestyle may be cumulative across a variety of activities, not all of which will be measured in a given cohort. In the case of medications and disease, while we account for age, sex, race, season, and levels of related disease biomarkers when examining the associations between medication use and individual genera, these adjustments may not fully disentangle the impact of medications on the microbiome from that of the associated disease condition. Further experimental studies using in vitro or in vivo models are needed to better understand the direct impact of specific medications on gut microbes independent of disease state. Regardless of these important caveats, it is clear that the microbiome is intimately connected to a wide range of host phenotypes and the ability to engineer our commensal microbiota will likely be an important component of precision medicine in the 21st century.

## Methods

**IRB approval for the study**. Procedures for the current study were run under the Western Institutional Review Board at Arivale Inc. (Seattle, WA) and at the Institute for Systems Biology (IRB Study Number 1178906 and 20170658, respectively).

**Data used in the study**. All de-identified data are from consenting individuals in a commercially-available, non-medical lifestyle intervention program (Arivale Inc., Seattle, WA). Only data from individuals who provided explicit informed consent and authorization for their anonymized data for research use are included in this study. Data were collected starting in July 2015. The Arivale program involved health coaching on exercise, nutrition, stress management, and other wellness goals. Only baseline data prior to intervention from a total of 3409 individuals who reported not taking antibiotics 3 months prior to sampling were included in the current study. This research project was performed independent of Arivale's commercial operations and was done entirely using only de-identified data.

**Clinical laboratory tests**. Participants' clinical blood laboratory tests were collected at LabCorp (North Carolina, USA) or Quest Diagnostics (Secaucus, NJ) facilities in the near vicinity of participants' geographic locations. Participants were directed to avoid alcohol, vigorous exercise, aspartame, and monosodium glutamate for a 24 h period prior to the blood draw, and to begin fasting 12 h in advance. Weight, height, waist circumference, and blood pressure measurements were collected at the time of each blood draw. Clinical blood laboratory tests involved a lipid panel and complete blood cell counts, as well as markers of diabetes, inflammation, liver function, kidney function, and nutrition (Supplementary Data 1).

**Gut microbiome sequencing**. Fresh stool specimens were taken at participants' homes using a sterile spoon or swab and were immediately preserved using chemical DNA stabilizers (OMNIgene Gut) to maintain DNA integrity. Previous analyses have demonstrated the ability for these stabilizers to preserve samples sufficiently at varying temperature conditions over multiple weeks with results comparable with immediate freezing[61–63]. Microbial DNA was then isolated from 250 μL of homogenized stool, using an automated protocol and MoBio's Power-Mag® Soil DNA isolation kit (+ClearMag®) microbiome DNA isolation kit on the KingFisher™ Flex instrument. This extraction protocol involved a bead beating step for mechanical lysis using glass beads and plate shaker for recovery of more DNA from a more diverse microbial community, as previously recommended[64]. Concentrations of extracted DNA from each sample were determined by Qubit, and an estimate of sample purity was determined via spectrophotometry by measuring the A260/A280 absorbance ratio. Gut microbiome sequencing data in the form of FASTQ files were obtained based on either 250 bp paired-end MiSeq profiling of the 16S V4 region (Second Genome) or 300 bp paired-end MiSeq profiling of the 16S V3 + V4 regions (DNA Genotek).

**16S rRNA data analysis**. Operational Taxonomic Units (OTUs) read counts were calculated using the QIIME pipeline[65] (version 1.9.1; default parameters) with closed-reference OTU picking against the Greengenes database[66] (version 13_08). To account for differences in sequencing depth in diversity analyses, we rarified each sample to 50,000 reads, removing samples that had <50,000 reads from further analysis. Rarefaction curves showed a tendency toward saturation, revealing sufficient sequencing depth. To examine the functional capacity of participants' gut microbiome, we applied the PICRUSt[67] pipeline to predict the KEGG[68] orthology (KO) profile for each sample from the relative abundances of OTUs (see Methods).

We next applied MUSiCC[69] to these KO profiles to convert the relative abundances to the average copy numbers and aggregated the resulting MUSiCC-normalized KO profiles to pathways and modules based on the KEGG annotations.

**Validated assessments and self-reported data**. Baseline assessment data were collected using a web-based interface and timed to coincide with blood draws. Validated assessments include: Dietary Targets Monitor, Oxford Happiness Questionnaire (OHQ), International Personality Item Pool-50 (IPIP-50), and Perceived Stress Scale (PSS-4)[70–73]. We also developed a set of in-house questionnaires that were also administered to the participants to collect health history and lifestyle data: personal and family health history, lifestyle (e.g., sleep habits, physical activity frequency, smoking and alcohol use), and digestive health (e.g., frequency of bowel movements, diarrhea, laxatives). We performed formal psychometric analysis of assessment data focusing on the measures of reliability (defined here as internal consistency or general factor saturation) and factorial structures of OHQ, IPIP-50, PSS-4, as well as custom 4-question sleep quality and 30-question digestive health questionnaires. Analyses revealed reliable estimates consistent with those reported for the validated assessments previously: Cronbach's alpha was estimated at 0.80 for PSS-4, 0.92 for OHQ, ranged from 0.80 to 0.89 for the five IPIP-50 subscales. We established closed correspondence between the proposed and observed factorial structure (using exploratory and confirmatory factor analysis, CFA): a five-factorial structure for IPIP-50 and 1-factorial structures for OHQ and PSS-4. For the custom assessments, a one-factor structure was established for sleep quality and digestive health measures, and these also displayed high levels of internal consistency (0.86 and 0.70, respectively).

**Statistical analysis**. Statistical analysis was performed using R version 3.5.1. All correlations reported in the paper are based on Spearman's rank correlation coefficient unless noted otherwise. We adjusted for age, sex, race of the individual, the season in which the microbiome sample was taken, and the microbiome sequencing vendor prior to analyses. Multiple hypothesis correction for $p$ values was performed using the Benjamini–Hochberg method of False Discovery Rate (FDR) control[74].

**Alpha diversity**. We calculated various measures of alpha diversity using methods implemented in the QIIME pipeline. The alpha diversity measures calculated using rarefied reads includes Shannon's diversity, Faith's Phylogenetic Diversity index, number of OTUs observed, and Pielou's evenness index.

**Analysis of Bacteroidetes-adjusted diversity**. To examine which taxa are associated with Shannon diversity when the abundance of Bacteroidetes is accounted for, Pearson correlations were computed between residuals of a cubic polynomial regression of Shannon diversity on relative abundance of Bacteroidetes. The residuals from the regression model are referred to as Bacteroidetes-adjusted diversity.

**Analysis of cluster-specific associations**. To examine interactions between overall microbiome composition and associations of microbial diversity or taxa abundance with host factors, each sample was defined as one of four possible taxonomic clusters. Samples with over 85% relative abundance of firmicutes were defined as the high-Firmicutes cluster (HF, 91 samples). Samples with Firmicutes relative abundance between 60–85% were defined as the Firmicutes cluster (F, 2095 samples). Samples with <60% Firmicutes where *Bacteroides* relative abundance was higher than twice the *Prevotella* abundance were defined as the *Bacteroides* cluster (B, 941 samples), and samples with <60% Firmicutes where *Bacteroides* relative abundance was lower than twice the *Prevotella* abundance were defined as the *Prevotella* cluster (P, 282 samples).

**Taxonomy-based ordination of microbiome profiles**. Different dimension-reduction and ordination techniques were applied to the taxonomic profiles obtained from the samples. These included the principal component analysis technique edgePCA[28], and principal coordinate analysis using weighted UniFrac distance[75] and genus-level Bray-Curtis distance. Since edge PCA does not require rarefied reads, we used the non-rarefied table.

**Analysis of associations with Shannon diversity**. For each analyte (e.g., lifestyle, diet, clinical test), associations were tested by fitting linear regression models of Shannon diversity on each analyte, adjusting for age, sex, race of the individual, the season in which the microbiome sample was taken, and the microbiome sequencing vendor. The FDR of the resulting tests were controlled for at level $\alpha = 0.05$ (see Statistical Methods, above).

**Analysis of associations with microbiome genera or pathways**. For each analyte (e.g., lifestyle, diet, clinical test), associations were tested by fitting generalized linear models of the microbiome feature on each analyte, adjusting for age, sex, race, the season in which the microbiome sample was taken, and the microbiome sequencing vendor. The model distribution depended on the type of microbiome

feature: for microbial genera, a logistic regression model was assumed for genera present in <75% of samples, otherwise a Poisson regression model was assumed; a linear regression model was assumed for microbial pathway analysis. The FDR of the resulting tests were controlled for at level $\alpha = 0.05$ (see Statistical Methods, above).

**Analysis of the impact of medication usage on the microbiome**. Each of the participants self-reported whether they are currently using one of three medication categories: cholesterol-lowering medications (e.g., statins), anti-hypertensive medications (e.g., valsartan), and blood-sugar regulating medications (e.g., metformin). In order to test for associations between microbial features (genera or pathways) and medication use, a Poisson regression model of genera abundance or linear regression model of pathway abundance, was fit on an indicator of medication use, adjusting for the age, sex, and race of the individual, the samples' season, the microbiome sequencing vendor, and biomarkers relevant to the medication. Specifically, for cholesterol-lowering medications the models were further adjusted for the levels of LDL cholesterol; for anti-hypertensive medications the models were further adjusted for systolic and diastolic blood pressures measures; and for blood-sugar regulating medications the models were further adjusted for fasting glucose and insulin levels. The FDR of the resulting tests were controlled for at level $\alpha = 0.05$ (see Statistical Methods, above).

**Reporting summary**. Further information on research design is available in the Nature Research Reporting Summary linked to this article.

## Data availability

The Institute for Systems Biology manages data requests for non-profit research purposes and will grant access to qualified researchers. Data requests should be sent to: A.M. (andrew.magis@isbscience.org). Raw microbiome data has been previously made available under SRA accession SRP148278 [https://www.ncbi.nlm.nih.gov/sra/?term=SRP148278]. OTU picking used the Greengenes database (version 13_08, https://greengenes.secondgenome.com/). KEGG database was downloaded on 7/3/2017 (https://www.kegg.jp/kegg/download/).

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

## Acknowledgements

We are grateful for the participants in the Arivale wellness programs who agreed for their de-identified data to be used in this research. We also thank Jessica Citronberg, Elisa Sheng, Christopher Bare for helpful discussions and for providing comments on the data analysis and paper.

## Author contributions

O.M. designed the study, analyzed the data, and drafted the initial paper. O.M. and C.L.D. edited, revised, and updated the paper with additional analysis. S.A.K., B.S., J.C.L., N.D.P., S.M.G., and A.T.M. participated in interpreting the results, provided feedback, and approved the final paper.

## Competing interests

O.M., C.L.D., S.A.K., B.S., J.C.L., and A.T.M. were all former employees and shareholders of Arivale. N.D.P. was a former shareholder of Arivale. Arivale is no longer a commercially operating company as of April 2019. The remaining author declares no competing interests.
