## [Peer Review File · Nature Communications]

Reviewers' comments:

Reviewer #1 (Remarks to the Author):

The manuscript by Mahor et al describes the phylogenetic analysis of fecal samples of over 3000 subjects who participated in an US wellness program. A large set of metadata were collected of these subjects and used to describe associations. The most remarkable observation was the correlation of a potential diversity optimum on the Firmicutes-to-Bacteroidetes axis, replicating some earlier observed phenomenon.

Many observational studies like this have been reported, some of which included up to several thousands of subjects such as the studies of the Raes and Wijmenga groups as reported individually and by Falony et al Science 2016. While it is of interest to compare and contrast the findings of previous studies with new cohorts, the present manuscript seem to suffer from a series of methodological and other confounders that are summarized below.

Major points:

1. It is indicated that several of the characteristics of the participants were self-reported (see line 59) – it is not clear what the extent of self-reporting has been, what measures have been taken as to validate the self-reporting – external validation of all used metadata is essential and non-validated data should not be included.
2. Stool samples were taken at participants' homes – two self-collection kits were used: what was the difference between these kits and how were the samples transported to the site where the DNA extraction took place. It should also be described what effect the different kits had on the same samples and how these differed from the results obtained with the golden standard of immediate freezing of the samples and storage at -80C. It is well known that differential storage of samples may affect notably the abundance of Bacteroides spp and this may confound the observations
3. DNA was isolated using an automated protocol and the 'MoBio's PowerMag® (+ClearMag®) microbiome DNA isolation kit on the KingFisher 434 TM Flex instrument'. Was this step including a mechanical lysis step as recommended t/o the literature ? The authors should compare this method with the preferred method described by the Costea et al Nat Biotech 2017 study.
4. In several cases the phylogenetic data are clustered into enterotypes are used – however, this seems not to be according to the general consensus of enterotypes as described and discussed by Costea et al Nat Microbiol 2018.
5. The QIIME pipeline version 1.9.1 was used to analyze the data. This is an old version and the use of QIIME 2 is recommended – see Bolyen et al Nat Biotech 2019.

Few other points:

1. It was noted that the wellness program involved a company that seem to employ the first author. The role of the wellness company should be indicated.
2. All data should be deposited at an accessible repository – this seems not to be the case.
3. What medical ethical permission was given for this study – were the ICMJE guidelines followed and if so why not ?

Reviewer #2 (Remarks to the Author):

Review of "Health and disease markers correlate with gut microbiome across thousands of people"

The study by Manor et al, describes the microbiota of thousands of individuals taking part in a wellness program. They use metadata collected from the participants to identify correlations between these metadata and microbiota diversity and composition.

Major Points:

1. It is not clear how diversity was calculated based on the methods. Were the number of reads rarefied? How many reads per sample was collected? Was data rarefied for PCA analysis? A more detailed methods is needed for these data.
2. All microbiota diversity was calculated using Shannon index, which takes into account both richness and evenness. It would be helpful to see what exactly is driving the diversity differences by separating these 2 aspects of diversity. Are the firmicutes dominated microbiota more diverse because the composition is more evenly distributed or does this microbiota harbor more taxa than the bacteroides dominated microbiota, or perhaps both? It would also be helpful to query phylogenetic diversity between the groups, ie by calculating phylogenetic distances between taxa that make up a microbiota.
3. It is not clear how the authors can determine effects of what I am assuming is highly co-varying metadata. For example, they state that consumption of fruits and vegetables, duration of exercise, consumption of sugary beverages, and bowel health correlate with microbiota diversity. It seems highly likely that these variables co-vary, ie the exercisers are also the vegetable and fruit consumers etc. Is it possible to test these factors independently using these data. For example, are there enough exercisers that don't consume a lot of fruits and vegetables or maybe consume sugary beverages to clearly isolate the exercise variable? If not, it is important to state that these differences in diversity could be a result of cumulative lifestyle differences.
4. The use of the term "enterotypes" is problematic given the prior use of that term in relation to specific microbiota configurations in Arumugam et al, 2011, which may be confusing to readers. It may be helpful to compare the configurations you describe here with the ones identified in the prior study, which looks like there would be overlap, and also use another term.
5. The specific taxa identified in Figure 5 appear to also include those that were no longer relevant after adjusting for diversity. It would be more helpful to graph those that remained significant after the diversity correction, which is mentioned in the text but not clearly delineated in the figure (or elsewhere).
6. The inferred pathway calculation could also benefit from a diversity correction. The glycan and carbohydrate utilization being associated with the disease cluster is not surprising given that this cluster is Bacteroidetes dominated, a phyla known to encode a vast array of cazymes. At the very least it is important to point out that due to differences in the phyla level composition of the healthy v. diseased groups some of the differences in inferred functionality may simply be a result of this.
7. The section on effects of medication on the microbiota is problematic. The very fact that individuals are taking medication means that they are treating some condition and it is possible that it is the condition that is related to the microbiota configuration and not the disease. It is difficult to disentangle these variables in this type of study therefore it is incredibly important that the authors state possible confounders when making these types of comparisons and concluding that a specific factor is impacting the microbiota.
8. In the discussion section, the authors mention follow-up and longitudinal studies as helping to disentangle lifestyle factors but, in fact, these studies are problematic due to co-varying factors. Ideally randomized controlled interventional studies are the best way to tease apart effects of individual factors on the microbiota. It would be great if microbiota researchers could learn from the mistakes made by nutrition scientists (often a result of funding constraints) that relied heavily on longitudinal studies and thus are open to confusion stemming from studies like the one published recently concluding that red and processed meat is safe for consumption, flying in the face of decades of longitudinal and some interventional data. Their results were a result of using "GRADE"

scoring of studies that ranks observational studies quite low and randomized controlled interventional studies high. Since there were few "high ranking" interventional studies, the results contradicted long-standing conclusions making it seem like dietary recommendation on red and processed meat consumption might be false. If the authors are going to propose studies to disentangle effects on the microbiota they should propose those that will actually be the most helpful.

Minor Points:

1. Figure S3 and S4 are inverted on their y axis relative to Figure 2. Would be helpful to keep these consistent.
2. In Figures 5 and 6 it would be nice to clearly delineate the two groups mentioned in the text, ie "healthy" "non-healthy" on the y-axis.

Reviewer #1:

“The manuscript by Manor et al describes the phylogenetic analysis of fecal samples of over 3000 subjects who participated in an US wellness program. A large set of metadata were collected of these subjects and used to describe associations. The most remarkable observation was the correlation of a potential diversity optimum on the Firmicutes-to-Bacteroidetes axis, replicating some earlier observed phenomenon.

Many observational studies like this have been reported, some of which included up to several thousands of subjects such as the studies of the Raes and Wijmenga groups as reported individually and by Falony et al Science 2016. While it is of interest to compare and contrast the findings of previous studies with new cohorts, the present manuscript seems to suffer from a series of methodological and other confounders that are summarized below.”

We thank the reviewers for constructive criticisms and valuable suggestions. We have carefully addressed the specific comments raised by the reviewer regarding our manuscript as described below.

Major Points

1. It is indicated that several of the characteristics of the participants were self-reported (see line 59) – it is not clear what the extent of self-reporting has been, what measures have been taken as to validate the self-reporting – external validation of all used metadata is essential and non-validated data should not be included.

We acknowledge the reviewer’s concern and apologize for not being more specific. Many of our self-reported data were collected via validated assessments, specifically the Dietary Targets Monitor, Oxford Happiness Questionnaire, International Personality Item Pool-50, and Perceived Stress Scale. Other self-reported data are related to health history (i.e. personal health history and family health history), current lifestyle habits, and digestive health, consistent with previous population-based studies and clinical studies. Although it was not the goal of the current study to validate these questionnaires, we performed formal evaluation of the psychometric qualities of the collected assessment data as part of a larger effort. We have edited the manuscript to include the following text to provide more detail and clarification regarding our collection of self-reported data.

(Page 16, lines 425-443) *Baseline assessment data were collected using a web-based interface and timed to coincide with blood draws. Validated assessments include: Dietary Targets Monitor (DTM), Oxford Happiness Questionnaire (OHQ), International Personality Item Pool-50 (IPIP-50), and Perceived Stress Scale (PSS-4). We also developed a set of in-house questionnaires that were also administered to the participants to collect health history and lifestyle data: personal and family health history, lifestyle (e.g., sleep habits, physical activity frequency, smoking and alcohol use), and digestive health (e.g., frequency of bowel movements, diarrhea, laxatives). We performed formal psychometric analysis of assessment data focusing on the measures of reliability (defined here as internal consistency or general factor saturation) and factorial structures of OHQ, IPIP-50, PSS-4, as well as custom 4-question sleep quality and 30-question digestive health questionnaires. Analyses revealed reliable estimates consistent with those reported for the validated assessments previously: Cronbach’s alpha was estimated at 0.80 for PSS-4, 0.92 for OHQ, ranged from 0.80 to 0.89 for the five IPIP-50 subscales. We established closed correspondence between the proposed and observed factorial structure (using exploratory and confirmatory factor analysis, CFA): a five-factorial structure for IPIP-50 and 1-factorial structures for OHQ and PSS-4. For the custom assessments, a one-factor*

structure was established for sleep quality and digestive health measures, and these also displayed high levels of internal consistency (0.86 and 0.70, respectively).

2. Stool samples were taken at participants' homes – two self-collection kits were used: what was the difference between these kits and how were the samples transported to the site where the DNA extraction took place. It should also be described what effect the different kits had on the same samples and how these differed from the results obtained with the golden standard of immediate freezing of the samples and storage at -80C. It is well known that differential storage of samples may affect notably the abundance of *Bacteroides* spp and this may confound the observations.

Fresh stool samples were collected by the participants using either a sterile scooping spoon or sterile swab. Samples were immediately placed into a stabilizer solution (OMNIgene GUT) to maintain DNA integrity at varying temperatures following collection and then mailed to respective sequencing facilities. Previous analyses have demonstrated the ability of this stabilizing solution to preserve samples sufficiently under all temperature conditions over multiple weeks with results comparable with immediate freezing (Song et al. 2016 mSystems, Wang et al. 2018 Front. Cell. Infect. Microbiol, and Szopinska et al. 2018 BMC Microbiology). Previous studies (e.g. Mendes-Soares et al. 2019 JAMA Network Open) have also implemented similar protocols with respect to self-collection kits. We also took additional computational steps to address potential confounding factors, processing all samples through the same standardized bioinformatic pipeline. We focused our analyses to samples with greater than 50,000 reads and analyzed only abundant taxa at each taxonomic level (genus and above). Finally, we also added the microbiome vendor used as a covariate to all our models to statistically adjust for any difference between the two sequencing kits/providers. In doing so, we believe that differences due to the two different kits/vendors are minimized and associations are biologically meaningful. We have updated the manuscript to reflect this information:

(Page 15, lines 398-403) *Fresh stool specimens were taken at participants' homes using a sterile spoon or swab and were immediately preserved using chemical DNA stabilizers (OMNIgene Gut) to maintain DNA integrity. Previous analyses have demonstrated the ability for these stabilizers to preserve samples sufficiently at varying temperature conditions over multiple weeks with results comparable with immediate freezing.*

3. DNA was isolated using an automated protocol and the 'MoBio's PowerMag® (+ClearMag®) microbiome DNA isolation kit on the KingFisher 434 TM Flex instrument'. Was this step including a mechanical lysis step as recommended t/o the literature ? The authors should compare this method with the preferred method described by the Costea et al Nat Biotech 2017 study.

We thank the reviewer for bringing up the value of mechanical lysis in DNA extraction. The process by which we extracted the DNA does in fact include a mechanical lysis step as part of the standard protocol. The standard PowerMag protocol includes both mechanical lysing via bead beating with the PowerMag beads and chemical lysing by the PowerMag lysis solution. Thus, our method for DNA extraction was consistent with the preferred method described by Costea et al. (2017) in Nat Biotech. We have updated the methods section to include a discussion on mechanical lysis.

(Page 15-16, lines 405-408) *This extraction protocol involved a bead beating step for mechanical lysis using glass beads and plate shaker for recovery of more DNA from a more diverse microbial community, as previously recommended.*

4. In several cases the phylogenetic data are clustered into enterotypes are used – however, this seems not to be according to the general consensus of enterotypes as described and discussed by Costea et al Nat Microbiol 2018.

We apologize for misusing the term ‘enterotype’ and agree with the reviewer that it is a fraught term. We have thus revised the language in our manuscript. We recognize that the original term as defined by Arumugam et al, 2011, is misleading and have been challenged by other studies (e.g. Koren et al. 2013 Plos Comp Bio, Gorvitovskaia et al. 2016 Microbiome). In fact, we didn’t observe distinct taxonomic clusters but rather a continuum of compositions ranging from very low levels of overall Bacteroidetes, to high levels of either *Prevotella* or *Bacteroides*. We have thus changed the language throughout the manuscript to now use the term ‘cluster’ as it more accurately reflects the taxonomic groupings.

5. “The QIIME pipeline version 1.9.1 was used to analyze the data. This is an old version and the use of QIIME 2 is recommended – see Bolyen et al Nat Biotech 2019.”

We acknowledge that QIIME 2 is currently the recommended pipeline for 16S data analysis, but note that QIIME 1.9.1 is still a validated method with over 19,000 citations. We developed and standardized our bioinformatics analysis method prior to the publication of the QIIME 2 paper when QIIME 1.9.1 was the standard. In doing so, we sought to make our results comparable with previous studies, which also used QIIME 1 and its sub-components. Furthermore, while analysis with amplicon sequence variant (ASV) has greater specificity than OTUs, we note that we are aggregating and analyzing on the genus level and above, minimizing differences between the two approaches. As such, while we agree that QIIME 2 is the recommended method for any future work we plan to do, we believe that existing and previous analyses using QIIME 1 are still valid.

Minor Points

1. “It was noted that the wellness program involved a company that seem to employ the first author. The role of the wellness company should be indicated.”

We apologize for our oversight on this and have updated the competing interest section of the manuscript. Commercial operations for Arivale have ceased as of April 2019. While certain authors were formerly employed by Arivale, no authors have been employed by Arivale since April 2019. As such, there is no active competing interest.

OM, CLD, SAK, BS, JCL, and ATM were all former employees and shareholders of Arivale. NDP was a former shareholder of Arivale. Arivale is no longer a commercially operating company as of April 2019.

2. “All data should be deposited at an accessible repository – this seems not to be the case.”

We agree that data should be made accessible for other researchers. We have implemented various approaches to enable access to the data and we have added the following addition to our manuscript:

(Page 20, lines 524-530) *Raw microbiome sequencing data has been previously made available under SRA accession SRP148278. Qualified researchers can access the full deidentified dataset for research purposes. Requests should be sent to the senior author: andrew.magis@systemsbiology.org. To protect the privacy of its participants and prevent reidentification, Arivale also encourages qualified*

researchers to obtain a license to data generated by Arivale's commercial service during its operation by contacting the current administrators of Arivale.

3. What medical ethical permission was given for this study – were the ICMJE guidelines followed and if so why not?

This study is a non-clinical, population-level, de-identified wellness study approved by Western Institutional Review Board (IRB Study Number 1178906 for Arivale and 20170658 for the Institute for Systems Biology). Participants of this study were not part of a registered clinical trial nor were clinical diagnoses provided to them. This research was performed independent of Arivale's commercial operations and was done entirely using de-identified data from individuals who had provided informed consent and research authorization for the use of their anonymized data in research, consistent with the approved IRB protocols:

(Page 15, lines 380-388) *All de-identified data are from consenting individuals in a commercially-available, non-medical lifestyle intervention program (Arivale Inc., Seattle, WA). Only data from individuals who provided explicit informed consent and authorization for their anonymized data for research use are included in this study. Data was collected starting in July 2015. The Arivale program involved health coaching on exercise, nutrition, stress management, and other wellness goals. Only baseline data prior to intervention from a total of 3,409 individuals who reported not taking antibiotics 3 months prior to sampling were included in the current study. This research project was performed independent of Arivale's commercial operations and was done entirely using only de-identified data.*

Reviewer #2:

"The study by Manor et al, describes the microbiota of thousands of individuals taking part in a wellness program. They use metadata collected from the participants to identify correlations between these metadata and microbiota diversity and composition."

We thank the reviewer for providing excellent feedback and suggestions to improve our manuscript. We have carefully addressed the specific comments raised by the reviewer regarding our manuscript as described below.

Major Points

1. *"It is not clear how diversity was calculated based on the methods. Were the number of reads rarefied? How many reads per sample was collected? Was data rarefied for PCA analysis? A more detailed methods is needed for these data."*

We apologize for the lack of detail and agree with the reviewer that more information can be provided. We calculated alpha diversity using the methods implemented in QIIME, including Shannon diversity index, Faith's Phylogenetic Diversity index, number of OTUs observed, and Pielou's evenness index. Data were rarefied to 50,000 reads per sample prior to diversity calculations. We also excluded samples with low total sequencing depth (<50,000 reads). We note that edgePCA does not require rarefaction and therefore data was not rarefied for edge PCA analysis. We have expanded the methods section to include:

(Page 16, lines 416-419) *To account for differences in sequencing depth in diversity analyses, we rarefied each sample to 50,000 reads, removing samples that had fewer than 50,000 reads from further analysis. Rarefaction curves showed a tendency towards saturation, suggesting sufficient sequencing depth.*

(Page 17, lines 449-452) *Alpha diversity. We calculated various measures of alpha diversity using methods implemented in the QIIME pipeline. The alpha diversity measures calculated using rarefied reads includes Shannon diversity index, Faith's Phylogenetic Diversity index, number of OTUs observed, and Pielou's evenness index.*

(Page 18, lines 472-473) *Since edge PCA does not require rarefied reads, we used the non-rarefied table.*

2. All microbiota diversity was calculated using Shannon index, which takes into account both richness and evenness. It would be helpful to see what exactly is driving the diversity differences by separating these 2 aspects of diversity. Are the firmicutes dominated microbiota more diverse because the composition is more evenly distributed or does this microbiota harbor more taxa than the bacteroides dominated microbiota, or perhaps both? It would also be helpful to query phylogenetic diversity between the groups, ie by calculating phylogenetic distances between taxa that make up a microbiota.

We thank the reviewer for bringing up this interesting point. To address this question, we have now added the new Supplementary **Figure S1**, showing that species richness (measured by the number of OTUs observed per sample), species evenness (measured by Pielou's evenness index), and phylogenetic diversity (measured by Faith's Phylogenetic Diversity index) all display a trend similar to Shannon diversity across the Bacteroidetes-to-Firmicutes axis. Specifically, Spearman's correlation coefficients with Shannon diversity were $\rho=0.97$, $\rho=0.83$, and $\rho=0.86$, for Pielou's index, observed OTUs, and Faith's index, respectively.

3. It is not clear how the authors can determine effects of what I am assuming is highly co-varying metadata. For example, they state that consumption of fruits and vegetables, duration of exercise, consumption of sugary beverages, and bowel health correlate with microbiota diversity. It seems highly likely that these variables co-vary, ie the exercisers are also the vegetable and fruit consumers etc. Is it possible to test these factors independently using these data. For example, are there enough exercisers that don't consume a lot of fruits and vegetables or maybe consume sugary beverages to clearly isolate the exercise variable? If not, it is important to state that these differences in diversity could be a result of cumulative lifestyle differences.

We agree with the reviewer that many features are likely co-correlated. We note that we did perform statistical tests to evaluate certain features independent of other co-varying features, as the reviewer mentioned. In particular, we tested the relationship between the microbiome and physical activity independent of vegetable consumption, fruit consumption, and other diet-related metadata. Under the section "Multiple lifestyle and clinical factors are associated with gut microbiome diversity", we stated that the correlations between microbiome diversity and moderate physical activity or vigorous physical activity are highly significant, and remained significant after adjustment for the weekly intake of fruits, vegetables, whole grains, and sugary drinks. By adjusting for intake of fruit, vegetable, other aspects of diet (and additionally BMI), we tested only the independent relationship between physical activity and microbiome diversity, which was still significant.

Nevertheless, our analyses do reveal redundant metadata features that are co-correlated with the microbiome, these co-varying associations are an important first step and may help identify future avenues for exploration, as it can help with resolving interrelationships. However, we fully understand that statistical adjustment does not always properly correct for co-varying features. Furthermore, we note that our goal was not to examine the causal effect of a specific feature on the microbiome or vice versa. That is, we were not attempting to identify host-microbiome causality, which is undoubtedly important but challenging to infer

with observational data. We have adjusted the manuscript to acknowledge the reviewer's concern and state how the result could be due to cumulative differences:

(Page 14, lines 358-366) *While our extensive phenotyping of host factors and our large sample size enables us to have the statistical power to identify many host-microbiome associations, we note that these associations do not imply causality. Understanding directional and causal relationships will require randomized control studies and interventions, which can control for both known and potential hidden confounders. Similarly, while we adjust for many co-varying factors, we note that statistical adjustment cannot fully account for all confounders, as there are many factors that remain unmeasured. For example, certain health-related activities are manifestations of lifestyle factors but the effect of lifestyle may be cumulative across a variety of activities, not all of which will be measured in a given cohort.*

4. The use of the term “enterotypes” is problematic given the prior use of that term in relation to specific microbiota configurations in Arumugam et al, 2011, which may be confusing to readers. It may be helpful to compare the configurations you describe here with the ones identified in the prior study, which looks like there would be overlap, and also use another term.

We agree with the reviewer that the term ‘enterotype’ is not appropriate here and apologize for the confusion. We recognize that the original term as defined by Arumugam et al, 2011, is misleading and have been challenged by other studies (e.g. Koren et al. 2013 Plos Comp Bio, Gorvitovskaia et al. 2016 Microbiome). In fact, we didn't find distinct clusters but rather a continuum of compositions ranging from very low levels of overall Bacteroidetes, to high levels of either *Prevotella* or *Bacteroides*. As such, we have removed any mention of the term ‘enterotype’ in the manuscript and have shifted to using the term ‘cluster’ as it more accurately reflects results.

5. The specific taxa identified in Figure 5 appear to also include those that were no longer relevant after adjusting for diversity. It would be more helpful to graph those that remained significant after the diversity correction, which is mentioned in the text but not clearly delineated in the figure (or elsewhere).

We thank the reviewer for this excellent suggestion and have now added an additional Supplemental Figure that only includes taxa that show strong associations with lifestyle and clinical features after adjusting for microbiome diversity (new supplementary **Figure S7**).

6. The inferred pathway calculation could also benefit from a diversity correction. The glycan and carbohydrate utilization being associated with the disease cluster is not surprising given that this cluster is Bacteroidetes dominated, a phyla known to encode a vast array of cazymes. At the very least it is important to point out that due to differences in the phyla level composition of the healthy v. diseased groups some of the differences in inferred functionality may simply be a result of this.

We thank the reviewer for this comment. As the reviewer points out, the functional composition of the microbial community is an outcome of the different taxonomic units that make up the community. We have now modified **Figure 6** to include the same notation as **Figure 5**, indicating which associations between functional pathways and lifestyle and clinical measurements are still significant after adjusting for microbiome diversity.

7. The section on effects of medication on the microbiota is problematic. The very fact that individuals are taking medication means that they are treating some condition and it is possible that it is the condition that is related to the microbiota configuration and not the disease. It is difficult to disentangle these variables in this type of study therefore it is incredibly important that the authors state possible confounders when making these types of comparisons and concluding that a specific factor is impacting the microbiota.

We agree with the reviewer that medication usages are often associated with existing disease or predisposition to a disease. We note that we have made efforts to separate the impact of medications on the gut microbiome by adjusting for measurements related to a disease (e.g. accounting for LDL levels for cholesterol-lowering drugs, or blood pressure for hypertension drugs). In doing so, we sought to understand the independent effect of the medications, going beyond previous studies that only explore the impact of disease or medications on the microbiome independently. We acknowledge, however, that such statistical adjustment is not always sufficient for correcting these confounding effects. We have modified the manuscript to acknowledge this limitation:

(Pages 13-14, lines 366-372) *In the case of medications and disease, while we account for age, sex, race, season, and levels of related disease biomarkers when examining the associations between medication use and individual genera, these adjustments may not fully disentangle the impact of medications on the microbiome from that of the overall associated disease condition. Further experimental studies using in vitro or in vivo models are needed to better understand the direct impact of specific medications on gut microbes independent of disease state.*

We further note that disentangling the impact of common medications on the microbiome independently from the disease is challenging. It is difficult to get ethical approvals to perform a study where certain individuals with a disease diagnosis are provided a standard medication while another group is deprived of this standard treatment, or conversely, where healthy individuals are given a drug. Thus, our observational study of microbe-medication associations is purely exploratory at this point, and more interventional work is needed to better account for these complex interactions.

8. In the discussion section, the authors mention follow-up and longitudinal studies as helping to disentangle lifestyle factors but, in fact, these studies are problematic due to co-varying factors. Ideally randomized controlled interventional studies are the best way to tease apart effects of individual factors on the microbiota. It would be great if microbiota researchers could learn from the mistakes made by nutrition scientists (often a result of funding constraints) that relied heavily on longitudinal studies and thus are open to confusion stemming from studies like the one published recently concluding that red and processed meat is safe for consumption, flying in the face of decades of longitudinal and some interventional data. Their results were a result of using “GRADE” scoring of studies that ranks observational studies quite low and randomized controlled interventional studies high. Since there were few “high ranking” interventional studies, the results contradicted long-standing conclusions making it seem like dietary recommendation on red and processed meat consumption might be false. If the authors are going to propose studies to disentangle effects on the microbiota they should propose those that will actually be the most helpful.

We thank the reviewer for the insightful comment. We agree that correlative studies suffer from hidden confounding factors and the results should not be interpreted with equal weight as randomized control trials. Ultimately, correlation does not equal causation and many studies unfortunately do extrapolate their results to sound like it’s causal. We therefore have changed the language in our manuscript to acknowledge this limitation and advocate for randomized studies, as the reviewer suggests:

(Page 14, lines 358-362) *While our extensive phenotyping of host factors and our large sample size enables us to have the statistical power to identify many host-microbiome associations, we note that these associations do not imply causality. Understanding directional and causal relationships will require randomized control studies and interventions, which can control for both known and potential hidden confounders.*

Minor Points

1. "Figure S3 and S4 are inverted on their y axis relative to Figure 2. Would be helpful to keep these consistent."

We thank the reviewer for this suggestion and we have now flipped the y-axis in **Figures S3 and S4** to match **Figure 2**.

2. "In Figures 5 and 6 it would be nice to clearly delineate the two groups mentioned in the text, ie "healthy", "non-healthy" on the y-axis."

We thank the reviewer for this suggestion and we have now added a caption and delineation of the two mentioned groups in **Figures 5 and 6**.

We hope that having comprehensively addressed all of the above comments you shall now find our revised manuscript fit for publication in *Nature Communications*.

Sincerely,

Ohad Manor, Ph.D.
Seattle, Washington

REVIEWERS' COMMENTS:

Reviewer #1 (Remarks to the Author):

The present manuscript is very much improved and most questions have been satisfactorily addressed. One point remains and that is the availability of the data where the authors now write: 'Qualified researchers can access the full deidentified dataset for research purposes. Requests should be sent to the senior author: andrew.magis@systemsbiology.org. To protect the privacy of its participants and prevent reidentification, Arivale also encourages qualified researchers to obtain a license to data generated by Arivale's commercial service during its operation by contacting the current administrators of Arivale.'

However, they also indicate earlier that 'Commercial operations for Arivale have ceased as of April 2019. While certain authors were formerly employed by Arivale, no authors have been employed by Arivale since April 2019. As such, there is no active competing interest.'

So it seems to me that qualified researchers are asked to get a license to Arivale's commercial service but this service reportedly has ceased to exist ...so what does this mean? This point has to be solved to get full transparency.

Reviewer #2 (Remarks to the Author):

I have no further comments or concerns.

Reviewer #1:

“The present manuscript is very much improved and most questions have been satisfactorily addressed.”

We thank the reviewer for their positive words and acknowledgment of the improved manuscript.

“One point remains and that is the availability of the data where the authors now write:

‘Qualified researchers can access the full deidentified dataset for research purposes. Requests should be sent to the senior author: andrew.magis@systemsbiology.org. To protect the privacy of its participants and prevent reidentification, Arivale also encourages qualified researchers to obtain a license to data generated by Arivale’s commercial service during its operation by contacting the current administrators of Arivale. ‘

However, they also indicate earlier that ‘Commercial operations for Arivale have ceased as of April 2019. While certain authors were formerly employed by Arivale, no authors have been employed by Arivale since April 2019. As such, there is no active competing interest.’

So it seems to me that qualified researchers are asked to get a license to Arivale’s commercial service but this service reportedly has ceased to exist ...so what does this mean ? This point has to be solved to get full transparency..”

We thank the reviewer for bringing up this point to allow us to clarify and alleviate any concerns. To address the reviewer’s concern, the Institute for Systems Biology (ISB) will manage the data sharing. ISB has permission to share the data with researchers engaging in non-profit research through a data use agreement (please see **Appendix I** below). This achieves the critically important aspect of scientific verifiability while also honoring the substantial amount of private investment it took to generate these data. We have revised the language in the manuscript to state that researchers can access the full dataset for non-profit research purposes through a data access request to ISB. This statement is consistent with the “Data are available on request due to privacy or other restrictions” example as outlined in *Nature’s* “Data availability statements and data citations policy: guidance for authors”. Additionally, this statement is consistent with the statement in our previous publication with *Nature Biotechnology* (PMID: 31477923). We will fully comply with the data availability policy of all *Nature* journals.

“The Institute for Systems Biology manages data requests for non-profit research purposes and will grant access to qualified researchers. We encourage researchers to apply for data access. Data requests should be sent to: Andrew Magis (andrew.magis@isbscience.org). Raw microbiome data has been previously made available under SRA accession SRP148278.”